# The added value of family-centered rounds in the hospital setting: A systematic review of systematic reviews

**Josien M. Woldring**[1,2]*, **Marie Louise Luttik**[1], **Wolter Paans**[1,3], **Reinold O. B. Gans**[2]

**1** Research Group Nursing Diagnostics, School of Nursing, Hanze University of Applied Sciences, Groningen, The Netherlands, **2** Department of Internal Medicine, University Medical Center Groningen, University of Groningen, Groningen, The Netherlands, **3** Department of Critical Care, University Medical Centre Groningen, Groningen, The Netherlands

* j.m.woldring@pl.hanze.nl

**Data Availability Statement:** All relevant data are within the paper and its Supporting Information files.

## Abstract

### Background

Family engagement in care for adult inpatients may improve shared decision making in the hospital and the competence and preparedness of informal caregivers to take over the care at home. An important strategy to involve family members in hospital care processes is to include them in (ward) rounds, also called 'family-centered rounds'(FCRs).

### Objectives

Summarize the evidence regarding the added value of FCRs from the perspectives of patients, family, and healthcare professionals.

### Methods

A review protocol was registered a priori with PROSPERO (number CRD42022320915). The electronic databases PubMed, CINAHL, and PsycInfo were searched for English-written systematic reviews with a focus on FCRs. The results and methods were presented in line with the PRISMA guidelines, and the methodological quality of the included reviews was assessed using the adapted version of the AMSTAR tool.

### Results

Of the 207 initial records, four systematic reviews were identified covering a total of 67 single studies, mainly performed in critical and pediatric care. Added values of FCR were described at review level, with references to single studies. All four systematic reviews reported an improvement in satisfaction among patients, family, and healthcare professionals, whereby satisfaction is linked to improved communication and interaction, improved situational understanding, inclusion of family in the decision-making process, and improved relationships within the care situation.

**Funding:** The author(s) received no specific funding for this work.

**Competing interests:** The authors have declared that no competing interests exist.

## Conclusion

Although only limited research has been conducted on the value of FCRs in the adult non-critical care setting, and despite the existence of a variety of outcome measures, the results available from the pediatric and acute care setting are positive. The findings of the sole study in an adult non-critical patient population are in line with these results. Further research in adult non-critical care is required to verify its effects in this setting.

## Introduction

Substantial evidence indicates the importance of family support for patients' health and well-being [1, 2]. Supportive social relations are known to favorably impact the physical, psychological, and cognitive functioning of patients [2]. Family members are therefore a major source of support to patients during hospitalization and during the process of recovery at home after discharge. Conversely, illness and disease affect not only the individual patient but also the family [3, 4]. With the growing scarcity of resources in European healthcare systems, there is an increasing need for families to take on caregiving responsibilities.

During hospital admissions, important decisions are often made regarding patients' treatment and care that might have consequences for their care situation after discharge as well. Furthermore, in most current healthcare systems, the length of a hospital stay is shortening, resulting in patients frequently being discharged while not yet fully recovered [5]. As a result, the care situation post-discharge is more complex, often with greater demands on caregivers at home [6, 7]. Research indicates that caregivers' preparedness for the role of caregiving highly depends on the support that they have received from healthcare professionals in the hospital [8].

According to Park et al. (2018), there is a call for renewed and innovative care delivery models that incorporate a patient- and family-centered approach [9]. Patient- and family-centered care (PFCC), as described by the Institute for Patient- and Family-Centered Care (IPFCC), is an approach to care-planning and -delivery based on establishing a collaborative partnership between the healthcare team, patients, and their families. The IPFCC defines PFCC as follows: 'working with patients and families instead of simply doing things to them or for them'. PFCC assumes that patient–family–clinician partnerships benefit clinicians as well as patients and their families [10].

An important strategy to involve family members in hospital care processes may be to include them in (ward) rounds, in what are called family-centered rounds (FCRs) [11]. Rounds can be seen as part of the daily communication process in hospitals between clinicians, nurses, and patients, often also involving other healthcare staff [12]. The goal is to formulate and communicate a shared understanding of the day's care and treatment plan for inpatients. How rounds are named, how they are structured, and who participates vary by provider, specialty, hospital, and country [12]. FCRs in this context are defined as 'multidisciplinary rounds at the bedside in which the patient and family are involved in creating the care plan and evaluating the process' [13]. Participation in the discussion and decision-making process can also be part of family presence [14]. With the current shift in healthcare from almost exclusive professional care to more informal family care, family engagement in the care for adult patients during rounds in the hospital may improve the ability and preparedness of informal caregivers to take over care at home. The objective of this systematic review is to summarize the evidence regarding the added value of FCRs from the perspectives of patients, family, and healthcare

professionals. In this study we consider 'added value' as an umbrella term for concepts such as outcomes, benefits, and effects.

## Methods

### Design

To obtain an overview of all available information on the added value of FCRs, we decided to conduct a systematic review of systematic reviews. The review protocol was registered a priori with PROSPERO (registration number: CRD42022320915). The methodological recommendations of Smith et al. and the Reporting Items for Systematic Reviews and Meta-Analyses (PRISMA) guidelines were followed in conducting and presenting this systematic review of systematic reviews [15, 16]. See S1 Checklist, PRISMA checklist. In this study a systematic review is, as defined by Cochrane Handbook, a review of a clearly formulated question that uses explicit, systematic methods to identify, select, and critically appraise relevant research, and to collect and analyze data from the studies that are included in the review [17].

### Search strategy and eligibility

In April 2022, the electronic databases PubMed, CINAHL, and PsycInfo were searched for relevant systematic reviews. To obtain as much information as possible on the value of FCRs, we focused on all care settings within the hospital, including patients of all ages and both critical and non-critical care. The search strategy and eligibility criteria were developed with the support of a database search expert from the University Medical Center Groningen. The search strategies are presented per database in Table 1. Systematic reviews were eligible for inclusion if they (I) included family presence in rounds, (II) included outcomes focused on the added

**Table 1. Search strategy for systematic review by database.**

| Database | Search Strategy |
|---|---|
| PubMed | ("Teaching Rounds"[Mesh] OR round*[tiab]) <br> AND <br> ("Family"[Mesh] OR "Caregivers"[Mesh] OR "Family Nursing"[Mesh] OR famil*[tiab] OR caregiver*[tiab] OR care giver*[tiab]) <br> AND <br> ("Meta-Analysis" [Publication Type] OR "Systematic Review" [Publication Type] OR systematic review [tiab] OR metaanal*[tiab] OR meta-anal*[tiab] OR systematic[sb]) |
| CINAHL | ((MH "Patient Rounds") OR (TI round*) OR (AB round*)) <br> AND <br> ((MH "Family+") OR (MH "Family Nursing") OR (MH "Caregivers") OR (TI famil* OR "care giver*" OR caregiver*) OR (AB famil* OR "care giver*" OR caregiver*)) <br> AND <br> ((MH "Meta Analysis") OR (MH "Systematic Review") OR (TI "Systematic Review") OR (AB "Systematic Review") OR (TI metaanal*) OR (AB metaanal*) OR (TI meta-anal*) OR (AB meta-anal*)) |
| PsycInfo | (TI round* OR AB round*) <br> AND <br> ((DE "Family" OR DE "Biological Family" OR DE "Dual Careers" OR DE "Dysfunctional Family" OR DE "Extended Family" OR DE "Family Background" OR DE "Family History" OR DE "Family Members" OR DE "Family of Origin" OR DE "Family Relations" OR DE "Family Resemblance" OR DE "Family Structure" OR DE "Family Work Relationship" OR DE "Interethnic Family" OR DE "Interracial Family" OR DE "Military Families" OR DE "Nepotism" OR DE "Nuclear Family" OR DE "Schizophrenogenic Family" OR DE "Stepfamily" OR DE "Caregivers") OR TI (famil* OR "care giver*" OR caregiver*) OR AB (famil* OR "care giver*" OR caregiver*)) <br> AND <br> (DE "Systematic Review" OR TI "Systematic Review" OR AB "Systematic Review" OR TI metaanal* OR AB metaanal* OR TI meta-anal* OR AB meta-anal*) |

value of FCRs, and (III) were written in English. Supplementary searches were done by checking reference lists of individual studies and PROSPERO review protocols.

### Study selection

The selection of the systematic reviews and presentation of the results was carried out according to the PRISMA guidelines. First, the titles and abstracts of all identified unique records were read independently by the first and second authors (JMW and MLL) to exclude reviews that did not meet the inclusion criteria. Second, the full texts of the records to be included were independently reviewed by the same two authors (JMW and MLL). There were four records with no agreement. Disagreement was solved through discussion with a third author (WP).

### Quality appraisal

The adapted version of AMSTAR was used to assess the methodology of the included reviews [18–20]. The adaptions, as described by De Groot et al. [21], were in line with the recommendation of Burda et al. [20], and concerned improvements for the usability, reliability, and validity of the tool. Each review, independently assessed by two authors (JMW and MLL), received an individual score between 0 and 12. Scores of 0–4 on the adapted version of AMSTAR were classified as low quality, scores 5–8 as average quality, and scores 9–12 as high methodological quality. Discussion to reach consensus followed when there were discrepancies in the scores between the two authors.

### Data extraction and synthesis

Results are presented using a descriptive synthesis. The included systematic reviews were systematically and independently reviewed using the PRISMA review protocol [16]. The research team developed a table with variables to be extracted from the selected reviews in order to guide the data extraction process. Two reviewers (JMW and MLL) read each review carefully and extracted the following data into the table for comparison: authors, country, research question, target population, number of studies included, design of the included studies and the results regarding the added values of FCRs from the perspectives of patients, families, and healthcare professionals. Once the data were extracted and grouped, the research team discussed the data and synthesized categories based on the findings with regard to 'added values'.

## Results

Fig 1 illustrates the PRISMA flow diagram of the systematic review. The searches led to 207 records, after removing duplicates 148 remained. In total 135 records appeared to be not related to FCRs and were hence excluded. Based on the full-text assessments, three publications did not address family involvement in rounds, and another six did not address added value outcome of FCRs. These nine publications were excluded, ultimately resulting in four systematic reviews eligible for inclusion. See S1 Appendix Search Strategy.

### Quality assessment

The quality assessment revealed an average (score: 7–8) to high quality (score: 9) of the reviews (Table 2). None of the four reviews contained a list of excluded studies, and none of the reviews considered relevant subgroups. Gray literature was also excluded in all reviews. Additionally, the review of Cypress et al. [22] (AMSTAR: score 7) lacked an assessment of the risk of bias on the single study level and missed an assessment of the likelihood of publication bias.

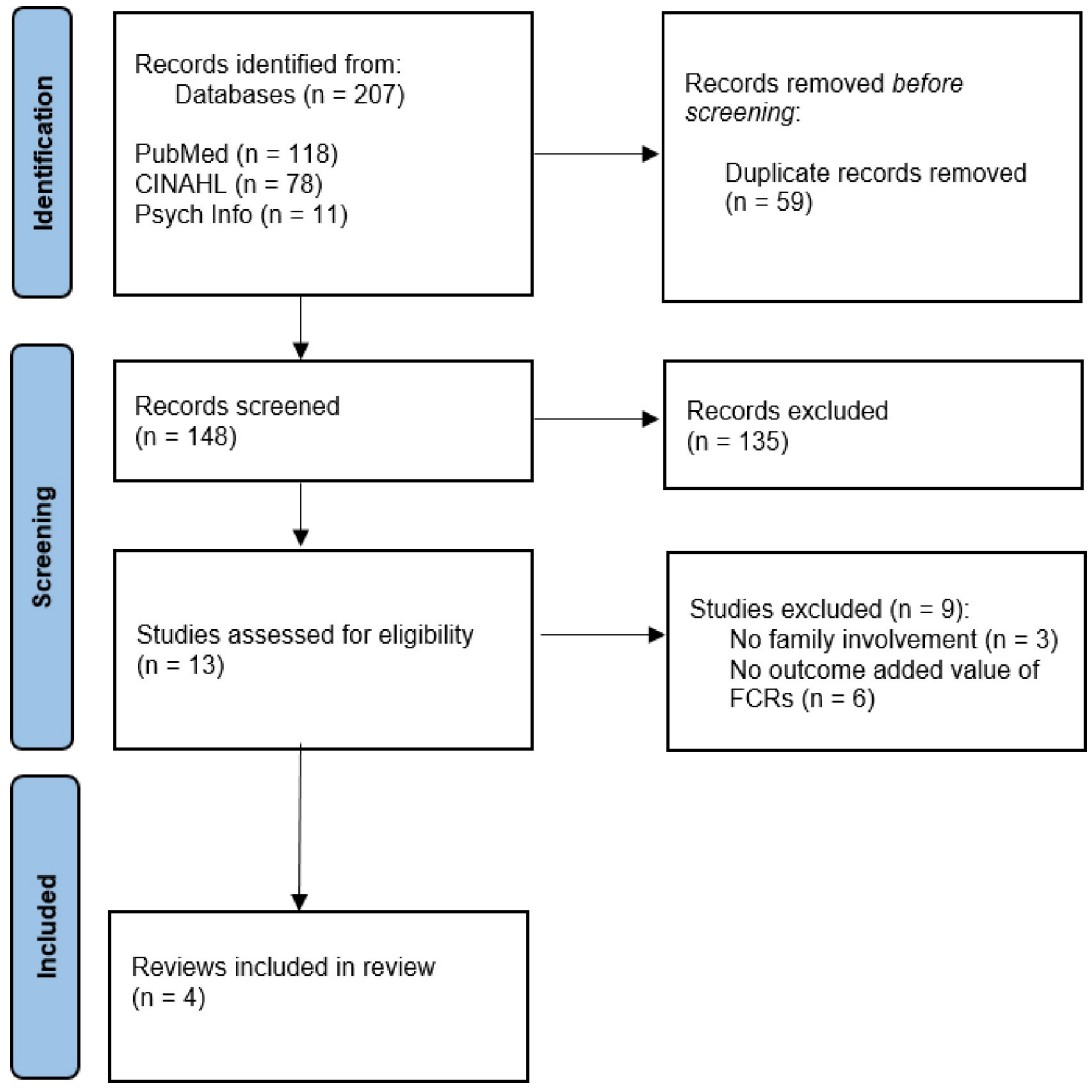

**Fig 1. PRISMA flow diagram for eligible article identification.**

Fernandes' review [23] (AMSTAR: score 8) did not appropriately assess the quality of the body of evidence.

## Study characteristics

Table 3 provides an overview of the main characteristics of the included reviews. The four reviews cover 67 single studies (91 in total, with 24 duplications), mainly performed in the United States and Canada. The included studies of the four reviews had different designs. Most evidence came from observational and survey studies, but RCTs were also included. Cypress et al. [22] included 19 single studies that were carried out in diverse settings ranging from adult to pediatric care and from critical to non-critical care. Two included studies focused on adult patients: one study in the ICU setting and another in an adult inpatient internal medicine department. All other included studies were conducted in the pediatric setting, with the reviews by Rea at al. [24] and Fernandes et al. [23] focusing specifically on this setting. The review of Rea et al. included 28 and the review of Fernandes et al. 29 single studies in

**Table 2. Methodological quality of the reviews included.**

| First author, (year of Publication) | Question 1[a] | Question 2[b] | Question 3[c] | Question 4[d] | Question 5[e] | Question 6[f] | Question 7[g] | Question 8[h] | Question 9[i] | Question 10[j] | Question 11[k] | Question 12[l] | Total AMSTAR score[j] | Methodological quality |
|---|---|---|---|---|---|---|---|---|---|---|---|---|---|---|
| Cypress (2012) | 1 | 1 | 0 | 1 | 0 | 1 | 0 | 1 | 1 | 0 | 1 | 0 | 7 | Average |
| Rea (2018) | 1 | 1 | 0 | 1 | 0 | 1 | 1 | 1 | 1 | 1 | 1 | 0 | 9 | High |
| Kydonaki (2021) | 1 | 1 | 0 | 1 | 0 | 1 | 1 | 1 | 1 | 1 | 1 | 0 | 9 | High |
| Fernandes (2021) | 1 | 1 | 0 | 1 | 0 | 1 | 1 | 0 | 1 | 1 | 1 | 0 | 8 | Average |

AMSTAR scores: 0 = no/1 = yes/c.a. = cannot answer.

[a]Question 1 = Were the review questions and inclusion/exclusion criteria clearly defined prior to executing the search strategy?

[b]Question 2 = Was a comprehensive literature search performed?

[c]Question 3 = Was relevant gray literature included in the review?

[d]Question 4 = Was there duplicate study selection and data extraction?

[e]Question 5 = Was a list of studies (included and excluded) provided?

[f]Question 6 = Were the characteristics of the included studies provided?

[g]Question 7 = Was the risk of bias assessed for each included study, taking into account important potential confounders and other sources of bias relevant to the review question?

[h]Question 8 = Was the quality of the body of evidence appropriately assessed and considered in formulating the conclusions of the review?

[i]Question 9 = Were the data appropriately synthesized in a qualitative manner and if applicable, was heterogeneity assessed?

[j]Question 10 = Was the likelihood of publication bias assessed?

[k]Question 11 = Were conflicts of interest disclosed for all of the review authors and was the funding source of the review and of each study within the review reported?

[l]Question 12 = Were relevant subgroups considered in the review process, analysis, and conclusions?

[j]Score: Low 0–4, Average 5–8, High 9–12

**Table 3. Characteristics of the included reviews.**

| First author, year of publication | Research question(s) | Outcomes | AMSTAR rating | Setting | Search years | Number of studies included | Countries | Design (number of studies) |
|---|---|---|---|---|---|---|---|---|
| Cypress, 2012 [22] | In critical and noncritical pediatric and adult patients, does family presence on rounds compared with non-inclusion of family members lead to positive outcomes and increased satisfaction? | Family members outcomes (positive and negative). Healthcare staff outcomes (positive and negative). | 7 | Pediatric & adult / critical care & non-critical care | 1988–2010 | 19 | USA (14) Canada (3) Israel (1) UK (1) | Randomized controlled trial (2) Quasi-experimental (1) Observational (12) Qualitative (1) Mixed methods (1) Quality improvement report (2) |
| Rea, 2018 [24] | What are families' experiences with the current model of FCR? How do families perceive FCR? What benefits or disadvantages do families see in FCR? | Overall parent experience. Impact of FCR on parent and family outcomes. Impact on FCR on parent psychosocial functioning. Parent relationships with medical teams. Pediatric patients' experiences with FCR. Barriers to FCR. | 9 | Pediatric | 2007–2017 | 28 | USA (23) Canada (3) Pakistan (1) New Zealand (1) | Randomized controlled trial (2) Quasi-experimental (2) Observational (2) Survey (13) Mixed methods (3) Quality improvement report (4) Focus Groups (2) |
| Kydonaki, 2021 [25] | How are family rounds implemented in adult critical care? What is the effect of family involvement in rounds in adult critical care for patients, family members and healthcare professionals? | Interactions and communication. Organization of rounds. Intensive care unit culture. | 9 | Adult & critical care | 1950–2019 | 15 | USA (10) Canada (5) | Pre- and posttest (4) Prospective-Parallel-group (1) Survey (6) Qualitative (3) Mixed methods (1) |
| Fernandes, 2021 [23] | Do FCRs promote humanistic pediatric care? | Empathy. Enhanced communication. Partnership. Respect. Satisfaction and service. | 8 | Pediatric | Journal inception—2020 | 29 | USA (24) Canada (2) Pakistan (1) Finland (1) France (1) | Randomized controlled trial (3) Pre- and posttest (7) Survey (11) Qualitative (3) Mixed methods (5) |

pediatric (non)- critical care with 14 duplicates between these two reviews [23, 24]. Furthermore, Kydonaki et al. [25] described family rounds in adult patients but with a specific focus on the critical care setting. Their review included 15 single studies; the majority (12) performed in the latter five years of the study.

The review of Fernandes et al. [23] mainly focused on advancing care from a humanistic frame of mind through FCRs. Therefore, this review gives a slightly different perspective of FCRs. On closer inspection, however, it appears that the intention with which the research was conducted is the same as the other reviews and this research provides an answer to the research question.

## Added value

The described added values of family rounds have been looked at the review level. The included reviews refer to single studies. Therefore, the references mentioned in these reviews are reproduced in Table 4. The contribution of this study is that synthesis is presented of the added values described in the four reviews from the patient's perspective, the perspective of the family, and the perspective of professionals.

**Satisfaction.** All four reviews reported that family presence during rounds seems to increase satisfaction among patients, their families, and healthcare professionals. However, none of the reviews present a clear definition of the concept of satisfaction. Satisfaction was explored in different study designs and with the use of numerous instruments. This shows an inconclusive and rather heterogeneous picture of the concept. Therefore, in this study satisfaction is operationalized toward specific aspects of satisfaction, such as *communication and interaction*, *situational understanding*, *inclusion in the decision-making process*, and *relationships* (Table 4).

**Added value for patients.** None of the four systematic reviews clearly reported on the added value from the patient's perspective. The concept of FCRs was studied primarily from the family perspective, with only three studies collecting data from the patient's perspective. The study by Rotman-Pikielny [26] included in Cypress' review [22] demonstrated that hospitalized adult patients in a general internal department would like their family members to participate in rounds and that participation contribute to a better understanding of their situation. In the review by Rea and colleagues [24], Berkwitt and Grossman interviewed pediatric patients and reported a wide variety of these participants' experiences [27]. The single study of Lewis et al. [28] confirmed this and noticed that children want to be able to hear rounds.

**Added value for family members.** *Improved communication and interaction for family members*. Including family in rounds has been found to improve communication and interaction between family members and healthcare professionals [28–36]. Family members look forward to having a specific time of the day to meet with healthcare professionals and the opportunity to communicate with them in simple language [26, 37–41]. Interaction during rounds makes it possible to share valuable and consistent information about a patient and provides an opportunity for a family to ask questions [26, 39, 42–47].

*Improved situational understanding for family members*. During FCRs, healthcare professionals share information about the patient, resulting in increased situational awareness and knowledge of the conditions and care (plan) among family members [31, 43, 45, 47, 48], who consequently better understand and appreciate the situation [30, 34, 37, 44, 49–54]. In the review of Kydonaki [25], the study of Cody and colleagues have also found that family members see FCRs as a roadmap for the patients' situation, goals, and expectations [45].

*Family members' inclusion in decision-making process*. All four reviews reported inclusion in decision making process, family members themselves welcomed the opportunity to offer input about the patient, advocate for the patient, ask questions, and be part of the discussion [29, 33, 55]. They have also been found to be supportive of the healthcare team in the decision-making process regarding future steps [26, 28, 34, 37, 40, 42, 56–58]. This involvement in decision-making can result in a sense of partnership between the family and healthcare professionals [59].

*Improved relationships between family members and healthcare providers*. The research indicates that due to improved interaction in FCRs, family feel included and respected by the healthcare team [28, 32, 34, 43, 60–63]. In some studies, family members were comfortable with the healthcare team, which increased confidence in the team and reduced stress [26, 29, 35, 42, 44, 48, 51, 64–66].

**Table 4. Description of added values.**

| First author, year of publication | Added value for patients | Added value for family members (FM) | Added value for healthcare professionals (HCP) |
|---|---|---|---|
| Cypress, 2012 [22] | | **Improved communication and interaction**<br>The opportunity to communicate with HCP [26, 37, 38].<br>Receiving information regarding the disease [26]. | **Improved communication and interaction**<br>Improved satisfaction with communication by the HCP team [33].<br>Decreased need for plan clarification by medical staff outside rounds [58]. |
| | **Improved understanding**<br>Contributed to a better understanding of their own situation [26]. | **Improved understanding**<br>Allowing better knowledge of the patient's condition [49].<br>Better understanding of the patient's care, conditions and plans [34, 37, 44]. | **Improved understanding**<br>HCP learned pertinent information from the family about the patient [38, 52]. |
| | | **Inclusion in the decision-making process**<br>Opportunity to give information, ask questions and be part of the discussion, involved in decision-making process [26, 28, 33, 37, 56]. | |
| | | **Improved relationship**<br>Increased feelings of inclusion and respect [34]. Positive impact on the attitude toward physicians [28].<br>Decreased family stress [26, 44]. | **Improved relationship**<br>HCP had a better ability to help families [33].<br>HCP had a greater sense of teamwork [33].<br>Improved staff's attitude towards the patient [26]. |
| Rea, 2018 [24] | | **Improved understanding**<br>Increased participation of FM, resulting in greater understanding about the patient's condition and treatment plan [50–54]. | **Improved understanding**<br>Medical team get new information from FM [50, 52, 64]. |
| | | **Inclusion in the decision making process**<br>Enhanced FM ability to advocate for the patient and impact the care plan [33, 55].<br>Greater inclusion in discussion a decision making [40]. | |
| | | **Improved relationship**<br>FM felt included [34, 60] which increased confidence in the medial team [51, 64, 65]. | |
| Kydonaki, 2021 [25] | | **Improved communication and interaction**<br>Improved interaction, communication with HCP [29–31].<br>Sharing valuable information and ask questions [39, 42–46]. | **Improved communication and interaction**<br>Improved interaction, communication and information sharing with FM [29, 42, 45, 46, 48].<br>Reduced number of meetings outside rounds [46]. |

(*Continued*)

Table 4. (Continued)

| First author, year of publication | Added value for patients | Added value for family members (FM) | Added value for healthcare professionals (HCP) |
|---|---|---|---|
| | | **Improved understanding**<br>Increased situational awareness [31, 43, 45, 48].<br>Better understanding of (care) goals, plans, expectations [30, 44–46, 48]. | **Improved understanding**<br>Improved understanding of the patient and family [30].<br>Increased HCP awareness of uncertain clinical situations [29, 43, 45].<br>Better understanding of FM gave HCP the opportunity to share treatment plan, and its related uncertainties with FM [43, 45]. |
| | | **Inclusion in the decision-making process**<br>FM as advocate for patient [29].<br>Included and supported in decision-making process, to make future steps [42, 57]. | |
| | | **Improved relationship**<br>Improved feelings of inclusion, being respected, and comfortable with physicians [43, 61].<br>Improved relationship and communication resulted in less stress and increased trust in the medical team [29, 42, 48]. | **Improved relationship**<br>Improved connection and relationship with FM [29]. |
| Fernandes, 2021 [23] | **Improved communication and interaction**<br>Children want to be able to hear rounds [28]. | **Improved communication and interaction**<br>Enhanced communication with HCP [28, 32–36].<br>Understanding the various team member roles [60].<br>HCP used simpler language [40, 41].<br>Perception of more consistent information [51].<br>Timeliness of questions answered [47]. | **Improved communication and interaction**<br>Improved communication with families [33, 35, 36, 67].<br>HCP are more likely to explain things clearly and spent more time [51]. |
| | | **Improved understanding**<br>Improvement of patient plan understanding [47]. | **Improved understanding**<br>HCP learned new information from FM [52].<br>FM input is helpful for HCP [60]. |
| | | **Inclusion in the decision-making process**<br>Increased involvement in decision-making [34, 37, 58] resulted in a sense of partnership [59].<br>Inclusion increased the ability to advocate for the patient [55]. | |
| | | **Improved relationship**<br>Increased perception of respect and empathy, FM felt more valued [28, 32, 62, 63].<br>More confidence and intimacy with HCP [35, 66]. | **Improved relationship**<br>Improved feeling of partnerships with FM [36, 67].<br>Increase in empathy and respect for FM [33, 51, 59, 62]. |

**Added value for healthcare professionals.** *Improved communication and interaction for healthcare professionals.* Improved communication and interaction with patients and their families has also been described as added values for healthcare professionals [29, 33, 35, 36, 42, 45, 48, 67]. This improvement enables the sharing of valuable information and explain things clearly [51], which has been found to a) decrease the need for plan clarifications later on and b) reduce the number of meetings outside rounds for healthcare professionals during a patient's hospital stay [46, 58].

*Improved understanding of the patient and family (situation).* Improved information sharing by healthcare professionals on a regular basis results in a better-informed family. This sharing affords healthcare professionals the opportunity to explain the treatment plan and its related uncertainties to family members [43, 45]. In some studies, healthcare professionals were also informed by family; they learned more about the patient's history, health, and life goals, thereby fostering an improved understanding of the patient and affecting the decision-making process [30, 38, 50, 52, 60, 64].

*Improved relationships between healthcare professionals, patients, and family members.* A better understanding of the patient and their family has been found to improve their mutual relationship [26, 29, 33, 51, 59, 62]. Healthcare professionals are consequently better able to help families and facilitate a sense of teamwork between healthcare professionals and family members [33, 36, 67].

## Discussion

This systematic review of systematic reviews provides an overview of the best available evidence regarding the added value of FCRs across different healthcare settings. The findings of our study are based on four systematic reviews, in which 67 separate studies were presented. The added value of FCRs was analyzed from the perspectives of patients, families, and healthcare professionals. Clear similarities were found between these three perspectives. The care situation in the setting of FCRs becomes transparent for all. From the perspectives of families and healthcare professionals, FCRs seem to improve satisfaction with communication, understanding of the care situation, and quality of the relationships. The purpose of FCRs–participation of the patient and family members in the discussion and decision-making process regarding patient care–seems to be achieved from the family perspective; however, it is not mentioned from the perspective of healthcare professionals. Family members also mention an increase in confidence and a decrease in stress related to the care situation.

The authors of the four systematic reviews concluded that the studies included in their review were too heterogeneous in their methodology and outcome measures, making it impossible to perform meta-analyses. Furthermore, it is remarkable that only two of the 67 included single studies were conducted in Europe. Most research was executed in the USA and Canada, which might suggest that FCRs are not yet widely adopted and does not seem a substantial subject for scientific studies outside the Angelo-Saxon countries.

FCRs have almost exclusively been studied from the perspective of families and/or healthcare professionals, but not patients. This is due to the fact that almost all studies have been conducted either in the pediatric (non-)critical care setting (51) or in the adult critical care setting (15). In the former group parents or guardians are the decision makers because of the young age of the patient population whereas in the critical care setting families are surrogate decision makers as patients are unresponsive most of the time.

In contrast, only one out of 67 studies included in the four reviews was conducted in an adult non-critical care setting, indicating that the study and adoption of FCRs in the latter setting have largely been ignored. This can be understood from the idea that children below 18

years of age are in most western countries by law not entitled to make decisions on their own, so parents or guardians are automatically involved, whereas in the critical care setting families are surrogate decision makers most of the time.

Family members are the major source of support for patients during hospitalization and at home. Therefore, family involvement is essential to ensure quality and continuity of care. Furthermore, in the current context of healthcare, with the growing number of care needs and increased complexity of care, the involvement of family caregivers becomes even more desirable and necessary. This is true for a wide variety of adult patients and not strictly for the most seriously ill, unresponsive ones or those without decision-making capacity [68].

The results derived from studies in the pediatric and critical care setting indicate added values for patients, families and healthcare professionals. Notably, the communication and interaction, situational understanding, inclusion in the decision-making process, and quality of relationships improved. Since the findings from the study performed in an adult internal department [26] seem to be in line with the studies performed in pediatric and critical care, FCRs could possibly also have a place in the adult non-critical care setting. However, the concept of FCRs will need re-examination and adjustment to a situation where communication moves from a bilateral character to communication in a triadic context where patients, family caregivers, and healthcare professionals are equal partners in discussion and decision-making.

The development of theories on how to involve family caregivers in care-planning and decision-making can be seen as a modern movement that fits society's general desire for its members to remain self-determining and independent of healthcare professionals for as long as possible [69]. FCRs provide the opportunity for patients to be involved in their own health decisions and for family members to be recognized as partners in care-planning, in decision-making, and in their role in providing comprehensive care at home [11, 68]. Furthermore, FCRs can be seen as tools to prepare and support patients and their family members in developing optimal self-care and self-management to deal with their health challenges. We recommend future research to fully explore the added value of FCRs in the adult non-critical care setting from the perspectives of patients, families, and healthcare professionals.

## Limitations

This study focused on the added value of FCRs. FCRs is apparently a complex clinical activity to investigate with quantitative study designs. There is no standardized measure for assessing FCR outcomes; most studies are based on observational and qualitative methodologies with a variety of (self-created) surveys and conducted interviews. Clear conclusions regarding the benefits of FCRs should therefore be drawn with caution. Quantitative studies with clear and appropriate outcome measures are necessary to provide better and conclusive evidence of the contribution of FCRs to the improvement of outcomes and well-being among patients, families, and healthcare professionals.

No analyses were conducted to determine the possible downside of FCRs. While some concerns were noted in the four reviews, such as an increased length of rounds, teaching aspects and potential privacy issues, they were not the focus of the reviews. These concerns should be part of further studies. Furthermore, the organization and education of FCRs are important aspects of implementation and should be studied. Furthermore, the organization and education of FCRs are important aspects of implementation. A structure or protocol with clear roles of FCRs participants should be studied and developed [70, 71].

In this review, only studies published in English could be included, as resources for extensive translations of non-English studies were not available. Non-English-speaking populations or contexts may consequently not be fully represented in this review. Furthermore, the finding

that most studies were executed in the USA and Canada, limiting the generalizability of the findings to the context of these countries. Research is hence required in the European context to establish the feasibility and added value in this specific healthcare context.

## Conclusion

FCRs seem to improve satisfaction among patients, their family members, and healthcare professionals, whereby satisfaction is linked to improved communication and interaction, improved situational understanding, and improved relationships within the care situation. However, current evidence is largely based on studies in the pediatric or critical care setting. Since family is the major source of support for patients and current healthcare more and more relies on the support of family, FCRs seem to be highly relevant for adult non-critical care patients as well.

In particular, it could help these patients and their family members in developing optimal self-care and self-management to deal with their health challenges after discharge from the hospital. This review supports a move toward accepting patients and their families as equal partners in the healthcare team and hence accepting family members as active participants in care-planning and decision-making. Future research is recommended to explore the contribution of FCRs in the hospital setting with adult non-critical care patients and their family members.

## Supporting information

**S1 Checklist. PRISMA checklist.**
(DOCX)

**S1 Appendix. Search strategy.**
(DOCX)

## Author Contributions

**Conceptualization:** Josien M. Woldring, Marie Louise Luttik, Wolter Paans, Reinold O. B. Gans.

**Data curation:** Josien M. Woldring.

**Formal analysis:** Josien M. Woldring, Marie Louise Luttik, Wolter Paans.

**Methodology:** Josien M. Woldring, Marie Louise Luttik, Wolter Paans, Reinold O. B. Gans.

**Project administration:** Josien M. Woldring, Marie Louise Luttik, Wolter Paans.

**Resources:** Josien M. Woldring.

**Supervision:** Reinold O. B. Gans.

**Validation:** Josien M. Woldring, Marie Louise Luttik, Wolter Paans, Reinold O. B. Gans.

**Visualization:** Josien M. Woldring.

**Writing – original draft:** Josien M. Woldring, Marie Louise Luttik, Wolter Paans.

**Writing – review & editing:** Josien M. Woldring, Marie Louise Luttik, Wolter Paans, Reinold O. B. Gans.

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
