## [Decision Letter · Decision Letter 0]

28 Jul 2022

PONE-D-22-14357The added value of family-centered rounds in the hospital setting: A systematic review of systematic reviewsPLOS ONE

Dear Dr. Woldring,

Thank you for submitting your manuscript to PLOS ONE. After careful consideration, we feel that it has merit but does not fully meet PLOS ONE’s publication criteria as it currently stands. Therefore, we invite you to submit a revised version of the manuscript that addresses the points raised during the review process. As editor, and to be fully transparent, I struggled to identify reviewers for this manuscript and so I performed the review myself to make sure that the process wasn't delayed.  Overall, both myself and the reviewer thought that this was a well-conducted study and has generally been written to a high standard. Some of my comments are aiming to improve the manuscript, and many are relatively small requests for changes / amendments to correct typos and grammatical mistakes. The larger comments relate to a better explanation of the data extraction and synthesis process, and reframing some of the discussion so that it is more impactful (ie drawing out the international relevance) and engaging with discussion around paediatric involvement in decision making. I've marked this as a 'major revision' but this is just to give you enough time to make the changes; in practice it could have been a 'minor revision'.  [Start of editor comments]

**Methods**

Lines 107-108 you state that outcomes, benefits and effects were used in the single systematic reviews. Can you, in response to reviewers, state which reviews these were? In the manuscript, you should cite these reviews or change the text to be more generic, ie away from specific reviews. You may want to consider moving the definition of added value to the end of the introduction and incorporate into the objective where the concept is first introduced in the text.

Line 110 is missing punctuation after the [15.16] citations.

In the study selection please state the number of agreements/disagreements that required adjudication by a third author.

In the data extraction and synthesis section it’s unclear what the process for extraction and synthesis was, particularly “included systematic reviews were systematically and independently reviewed by two reviewers” (lines 149-150). What does ‘reviewed’ mean in the context of data extraction? Did you use any software to support data extraction (eg Microsoft Excel)? Was a particular data extraction tool used? If so, how was the tool created / extraction headings selected? How were disagreements between the two reviewers addressed at point of extraction? Which two authors performed the extraction and synthesis?

**Results**

Figure 1 uses an outdated version of the PRISMA diagram, please update it to the current version.

You state that out of 91 studies included in the four reviews, there were 24 duplications. Do you know why there were so few (relatively) duplicates? Eg did the reviews have different search criteria or research questions? (this may be more relevant to reflect upon in the discussion than results)

The sentence starting on line 197, “These reviews included 28…” is hard to follow, please rephrase.

Line 201, “performed in the past 4[sic] years” would perhaps be better phrased as “performed in the latter four years of the study. Note that 2014-2018 is a five year period (14, 15, 16, 17, 18) – either amend to a latter five years or change to 2015-2018, whichever is accurate.

There are typos in the results, please proof read the full paper though I’ve tried to spot some. Line 254, Kydonaki needs to be capitalised. Line 264, “these” should be “this” and the sentence should continue immediately after the previous sentence.

**Discussion**

The third paragraph starting on line 316, “The findings of our study…” would be better placed at the end of the first paragraph.

You say that meta analysis wasn’t possible, but how do you know when you didn’t assess the individual studies yourself? Do you mean a meta-analysis of the systematic reviews’ data? This should be made clearer, as well as the potential and/or uncertainty around whether a meta-analysis of individual studies would be possible. Also consider making it clear that the judgement of studies being highly heterogeneous wasn’t yours (as you didn’t assess this for individual studies) but rather from the reviews themselves.

You state that almost all studies were in paediatric settings or critical care settings; please state how many (as you’ve done for adult non-critical care settings).

Line 323, “1” should be “one” (please check manuscript for all numbers of ten or less).

Line 325, children being “under-age” appears colloquial. Perhaps just remove this and leave it as “cannot take decisions on their own”. I also think that you should engage with debate about whether children can or cannot make informed decisions about their care as you appear to uncritically accept the statement that they cannot take decisions on their own. There’s an increasing body of work examining paediatric shared decision making in a broader context, though I’m unsure whether this has yet reached person centred ward rounds (may be an opportunity to better suggest it). Your final paragraph before the limitations begins to touch on this but not specifically in relation to paediatric involvement.

Line 334-335, please rephrase “highlighted positively” as it’s unclear.

Your final limitation, with only two of 67 studies being in Europe, arguably isn’t a limitation but rather one of the key findings from this review, that person centred ward rounds are yet to be properly studied in countries outside of the US. I think you should make more of this in your broader discussion rather than leaving it as a limitation. I understand it’s a grey area between being a finding and a limitation because only including papers written in English may have introduced this bias, but the general lack of English speaking countries (eg UK, Australia, Ireland) suggests that this is problem with either a) spread / adoption, or b) terminology (are there different terms used in other English speaking countries?). It would be good for you to consider and reflect upon these in the discussion, either earlier or if you decide to leave this in the limitations. 

[End of editor comments]

We look forward to receiving your revised manuscript.

Kind regards,

Jason Scott

Academic Editor

PLOS ONE

Journal Requirements:

Reviewers' comments:

Reviewer's Responses to Questions

**Comments to the Author**

1. Is the manuscript technically sound, and do the data support the conclusions?

Reviewer #1: Yes

2. Has the statistical analysis been performed appropriately and rigorously? 

Reviewer #1: N/A

3. Have the authors made all data underlying the findings in their manuscript fully available?

Reviewer #1: Yes

4. Is the manuscript presented in an intelligible fashion and written in standard English?

Reviewer #1: Yes

5. Review Comments to the Author

Reviewer #1: This is a well connected review and the conclusions reflect the data. The search strategy is particularly well reported and the quality audit of the reviews is easy to understand. The literature does support the call for more research, particularly in adult practice. An examination of the value of FCR in learning disability would be useful as these individuals rely on care givers in their efforts to communicate.

6. PLOS authors have the option to publish the peer review history of their article (what does this mean?). If published, this will include your full peer review and any attached files.

Reviewer #1: **Yes: **Peta Jane Greaves

---

## [Author Response · Author response to Decision Letter 0]

10 Sep 2022

Point of the editor and reviewer Response of the authors

Method 

Lines 107-108 you state that outcomes, benefits and effects were used in the single systematic reviews. Can you, in response to reviewers, state which reviews these were? In the manuscript, you should cite these reviews or change the text to be more generic, ie away from specific reviews. You may want to consider moving the definition of added value to the end of the introduction and incorporate into the objective where the concept is first introduced in the text.

 We agree with this comment, and your suggestion. We have moved the introduction of the concept of ‘added value’ to the end of the introduction.

Line 110 is missing punctuation after the [15.16] citations.

 Amended

In the study selection please state the number of agreements/disagreements that required adjudication by a third author.

 There were four records with no agreement between the two researchers. We have clarified this in the article. 

There were four records with no agreement. Disagreement was solved through discussion with a third author (WP). 

In the data extraction and synthesis section it’s unclear what the process for extraction and synthesis was, particularly “included systematic reviews were systematically and independently reviewed by two reviewers” (lines 149-150). What does ‘reviewed’ mean in the context of data extraction? Did you use any software to support data extraction (eg Microsoft Excel)? Was a particular data extraction tool used? If so, how was the tool created / extraction headings selected? How were disagreements between the two reviewers addressed at point of extraction? Which two authors performed the extraction and synthesis?

 Thank you for this comment. We have expanded the description of the extraction and synthesis process. 

The research team developed a table with variables to be extracted from the selected reviews in order to guide the data extraction process. Two reviewers (JMW and MLL) read each review carefully and extracted the following data into the table for comparison: authors, country, research question, target population, number of studies included, design of the included studies and the results regarding the added values of FCRs from the perspectives of patients, families, and healthcare professionals. Once the data were extracted and grouped, the research team discussed the data and synthesized categories based on the findings with regard to ‘added values’. Four different categories emerged: communication and interaction, situational understanding, inclusion of family in the discussion making process, and relationship within the care situation. 

Results 

Figure 1 uses an outdated version of the PRISMA diagram, please update it to the current version.

 The PRISMA diagram is updated to the current version. 

See figure 1 Flowchart

You state that out of 91 studies included in the four reviews, there were 24 duplications. Do you know why there were so few (relatively) duplicates? Eg did the reviews have different search criteria or research questions? (this may be more relevant to reflect upon in the discussion than results) The limited number of duplicates is mainly due to the fact that the reviews had different target populations (Rea and Fernandes focused on pediatrics and Kydonaki on adult critical care). The reviews of Rea et al. and Fernandes et al. have pediatrics as the target population and have most duplications (14). Kydonaki had the focus on adult critical care and therefore showed no duplications with Rea et al. and Fernandes et al. This is described in the study Characteristics.

The sentence starting on line 197, “These reviews included 28…” is hard to follow, please rephrase.

 The sentence is rephrased as: 

The review of Rea et al. included 28 and the review of Fernandes et al. 29 single studies in pediatric (non)-critical care with 14 duplicates between these two reviews [23,24].

Line 201, “performed in the past 4[sic] years” would perhaps be better phrased as “performed in the latter four years of the study. Note that 2014-2018 is a five year period (14, 15, 16, 17, 18) – either amend to a latter five years or change to 2015-2018, whichever is accurate.

 The sentence is rephrased as:

Their review included 15 single studies; the majority (12) performed in the latter five years of the study. 

There are typos in the results, please proof read the full paper though I’ve tried to spot some. 

• Line 254, Kydonaki needs to be capitalised. 

• Line 264, “these” should be “this” and the sentence should continue immediately after the previous sentence. The entire research team has read the paper again. 

Typos are amended

Discussion 

The third paragraph starting on line 316, “The findings of our study…” would be better placed at the end of the first paragraph.

 We agree and have replaced this line, as suggested, to the first paragraph.

You say that meta analysis wasn’t possible, but how do you know when you didn’t assess the individual studies yourself? Do you mean a meta-analysis of the systematic reviews’ data? This should be made clearer, as well as the potential and/or uncertainty around whether a meta-analysis of individual studies would be possible. Also consider making it clear that the judgement of studies being highly heterogeneous wasn’t yours (as you didn’t assess this for individual studies) but rather from the reviews themselves. 

 Amended to: 

The authors of the four systematic reviews concluded that the studies included in their review were too heterogeneous in their methodology and outcome measures, making it impossible to perform meta-analyses. 

You state that almost all studies were in paediatric settings or critical care settings; please state how many (as you’ve done for adult non-critical care settings).

 Amended to:

FCRs have almost exclusively been studied from the perspective of families and/or healthcare professionals, but not patients. This is due to the fact that almost all studies have been conducted either in the pediatric (non)-critical care setting (51) or in the adult critical care setting (15). In the former group parents or guardians are the decision makers because of the young age of the patient population whereas in the critical care setting families are surrogate decision makers as patients are unresponsive most of the time. 

Line 323, “1” should be “one” (please check manuscript for all numbers of ten or less).

 Amended

Line 325, children being “under-age” appears colloquial. Perhaps just remove this and leave it as “cannot take decisions on their own”. I also think that you should engage with debate about whether children can or cannot make informed decisions about their care as you appear to uncritically accept the statement that they cannot take decisions on their own. There’s an increasing body of work examining paediatric shared decision making in a broader context, though I’m unsure whether this has yet reached person centred ward rounds (may be an opportunity to better suggest it). Your final paragraph before the limitations begins to touch on this but not specifically in relation to paediatric involvement.

 Thank you for this important comment. We are aware of the debate whether children can or cannot make informed decisions. There exists no universal agreement as to at what age minors should be deemed decision-making competent, as this clearly depends on their maturity. As described in the final paragraph we see patient, family and healthcare professionals as equal partners in discussion and decision making. We described it broadly because, as described in the introduction, our focus is on the adult (non-critical care) patient. 

In this paragraph, we wanted to make clear that children below 18 years of age are in most western countries by law not entitled to make decisions on their own, but depending on their maturity may be increasingly involved from age 12 and beyond. 

Amended

This can be understood from the idea that children below 18 years of age are in most western countries by law not entitled to make decisions on their own, so parents or guardians are automatically involved, whereas in the critical care setting families are surrogate decision makers most of the time. 

Line 334-335, please rephrase “highlighted positively” as it’s unclear.

 Amended:

The results derived from studies in the pediatric and critical care setting indicate added values for patients, families and healthcare professionals.

Your final limitation, with only two of 67 studies being in Europe, arguably isn’t a limitation but rather one of the key findings from this review, that person centred ward rounds are yet to be properly studied in countries outside of the US. I think you should make more of this in your broader discussion rather than leaving it as a limitation. I understand it’s a grey area between being a finding and a limitation because only including papers written in English may have introduced this bias, but the general lack of English speaking countries (eg UK, Australia, Ireland) suggests that this is problem with either a) spread / adoption, or b) terminology (are there different terms used in other English speaking countries?). It would be good for you to consider and reflect upon these in the discussion, either earlier or if you decide to leave this in the limitations. 

 We agree that only two studies in Europe also is a finding from this review so we added the following in the discussion:

It is remarkable that only two of the 67 included single studies were conducted in Europe. Most research was executed in the USA and Canada, which might suggest that FCRs are not yet widely adopted and does not seem a substantial subject for scientific studies outside the Angelo-Saxon countries. 

We do not think there is a problem with terminology. We have done a broad search on rounds with Mesh terms etc. and found a lot of different rounds. It seems unlikely that we missed studies due to search terms. 

Limitations:

The finding that most research was executed in the USA and Canada, limiting the generalizability of the findings to the context of these countries. Research is hence required in the European context to establish the feasibility and added value in this specific healthcare context.

---

## [Decision Letter · Decision Letter 1]

2 Dec 2022

PONE-D-22-14357R1The added value of family-centered rounds in the hospital setting: A systematic review of systematic reviewsPLOS ONE

Dear Dr. Woldring,

Thank you for submitting your manuscript to PLOS ONE. After careful consideration, we feel that it has merit but does not fully meet PLOS ONE’s publication criteria as it currently stands. Therefore, we invite you to submit a revised version of the manuscript that addresses the points raised during the review process.

Thank you for the revision of your manuscript. Please find below the remarks and questions of the second reviewer. Can you please address them in order to proceed to acceptance?

We look forward to receiving your revised manuscript.

Kind regards,

Inge Roggen, M.D., Ph.D.

Academic Editor

PLOS ONE

Journal Requirements:

Reviewers' comments:

Reviewer's Responses to Questions

**Comments to the Author**

1. If the authors have adequately addressed your comments raised in a previous round of review and you feel that this manuscript is now acceptable for publication, you may indicate that here to bypass the “Comments to the Author” section, enter your conflict of interest statement in the “Confidential to Editor” section, and submit your "Accept" recommendation.

Reviewer #1: All comments have been addressed

Reviewer #2: (No Response)

2. Is the manuscript technically sound, and do the data support the conclusions?

Reviewer #1: Yes

Reviewer #2: Partly

3. Has the statistical analysis been performed appropriately and rigorously? 

Reviewer #1: N/A

Reviewer #2: N/A

4. Have the authors made all data underlying the findings in their manuscript fully available?

Reviewer #1: Yes

Reviewer #2: Yes

5. Is the manuscript presented in an intelligible fashion and written in standard English?

Reviewer #1: Yes

Reviewer #2: Yes

6. Review Comments to the Author

Reviewer #1: You have addressed my comments satisfactorily.I have no concerns about this publication. I note your comments about the international status of this topic

Reviewer #2: I would like to thank the editor and authors for this interesting paper on the added value of family-centered rounds in the hospital setting. This is an interesting topic with potential impact in clinical practice and for future research.

The design of the study seems sound. A review protocol was registered a priori and PRISMA guidelines were followed. The search and study selection strategy are explained in detail. To increase reliability, the authors selected studies independently and if there was disagreement a third author was involved. Quality appraisal was performed rigorously using the AMSTAR checklist which is used for appraisal of systematic reviews of randomised and non-randomised studies of healthcare interventions. Data extraction was done in a systematic manner. However, I still have some comments on the presentation of the results in this systematic review. My remarks can be found below.

1 - I was wondering, since the aim of this systematic review is to investigate the added value of family-centered rounds without clearly defining ‘added value’ a priori, this could be a scoping review rather than a classic systematic review. Scoping systematic reviews are done to identify key characteristics or factors related to a concept and it seems that this corresponds with the aim of this study. However, I am not entirely sure that it is a scoping review because this review summarised the effects of FCRs on patient, family member, and healthcare professional outcomes. However, I could not find the actual effect sizes and corresponding levels of uncertainty in this study which is common in a classic systematic review. Could the authors please explain which design they followed and substantiate their decision? Another remark is that PROSPERO registration is not allowed for scoping reviews which could be an additional issue if the authors change the design.

2 - I would remove the sentence: “Four different categories emerged: communication and interaction, situational understanding, inclusion of family in the discussion making process, and relationship within the care situation.” from the method section. These are results and therefore do not belong there.

3 - Please cite the excluded studies (n=9) in the second last step of the PRISMA flow chart. This allows the reader to review these excluded studies.

4 - Table 3: Is it possible to provide the total number of participants per included review? A separate column specifying the different outcomes studied would also be helpful. The ‘target population’ is in my opinion the ‘setting’ since the population could be healthcare workers, patients and/or family.

5 - In a systematic review of systematic reviews summary results should be presented per included systematic review in a summary of findings table. The authors stated that included studies were too heterogeneous in their methodology and outcome measures, making it impossible to perform meta-analyses. And I do agree that this is the case here. However, the authors could decide to report effect sizes and corresponding levels of uncertainty per type of outcome for each category (e.g., family member satisfaction) in a table. If necessary, the different definitions of outcomes could be provided next to the results. Moreover, the authors could decide to select studies with a certain level of evidence (e.g. RCTs and/or quasi-experimental research) and provide a table with the respective effect sizes per outcome measure. I think it would be interesting to present some numbers in this systematic review.

6 - p20. The authors state: “However, in the current context of healthcare, with the growing number of care needs and increased complexity of care, the involvement of family caregivers is desirable and necessary. Current healthcare systems are increasingly relying on family members to take on care responsibilities. This is also true for a wide variety of adult patients and not strictly for the most seriously ill, unresponsive ones or those without decision-making capacity [69].”

Yes, this is true. However, isn’t an increased reliance on family members because of shortages in professional caregivers a negative evolution? The authors should nuance their message here or at least provide some more explanation. I do not think that the argument is entirely correct, and this statement is repeated in the conclusion (In the light of the current challenges in healthcare, FCRs seem to be highly relevant for adult non-critical care patients as well.)

7 - An important limitation of this review is the limited focus on the added value of FCRs. The authors acknowledge this in the discussion and mention that implementation should be studied. However, could this be more substantiated with relevant literature?

7. PLOS authors have the option to publish the peer review history of their article (what does this mean?). If published, this will include your full peer review and any attached files.

Reviewer #1: **Yes: **Peta Jane Greaves

Reviewer #2: **Yes: **Filip Haegdorens

---

## [Author Response · Author response to Decision Letter 1]

20 Dec 2022

Dear reviewer,

Thank you for your careful review of our manuscript on the added values of family-centered rounds. 

We welcome your constructive criticism and have improved the manuscript accordingly. 

Below you find a response to each comment raised by the reviewer. 

1 - I was wondering, since the aim of this systematic review is to investigate the added value of family-centered rounds without clearly defining ‘added value’ a priori, this could be a scoping review rather than a classic systematic review. Scoping systematic reviews are done to identify key characteristics or factors related to a concept and it seems that this corresponds with the aim of this study. However, I am not entirely sure that it is a scoping review because this review summarised the effects of FCRs on patient, family member, and healthcare professional outcomes. However, I could not find the actual effect sizes and corresponding levels of uncertainty in this study which is common in a classic systematic review. Could the authors please explain which design they followed and substantiate their decision? Another remark is that PROSPERO registration is not allowed for scoping reviews which could be an additional issue if the authors change the design. 

The objective of this systematic review of systematic reviews is to summarize the evidence regarding the added value of FCRs from the perspectives of patients, family, and healthcare professionals. 

We agree with the reviewer that the concept of 'added value' is not unambiguously interpreted in the various studies included. We have considered presenting a certain one-dimensional definition in advance. Ultimately, we did not do this in order to do justice to the interpretations in the various studies. Nevertheless, we believe that the current methodical, systematic approach and the questioning justifies the interpretation of a systematic review. In our review it was not the main goal to discuss characteristics of the concept ‘added value’ (Indeed, in that case a scoping review could be more opportune). In our review a clinically meaningful question was chosen as a focus, in which a systematic review seems to be the best fitting approach. i.e. The added value of family-centered rounds in the hospital setting; not specifically the analysis of the conceptual specifications, antecedents or theoretical representations of ‘added value’.

Indeed, the most traditional form of a systematic review has not been chosen here, which only includes effect studies that refer to double-blind randomized studies and associated effect measures. However, we believe that the chosen diversity in included designs does not have to be a major objection to abandoning the systematic method. 

It was our choice to look for a synthesis at the level of ‘systematic reviews’ and not primarily at the independent studies included in it. In our opinion it is therefore more logical to stick to the 'systematic review' design.

We have therefore chosen to state the estimated level of evidence in the studies; something that also belongs to a systematic review.

We hope to have provided sufficient argumentation to continue with a systematic review.

2 - I would remove the sentence: “Four different categories emerged: communication and interaction, situational understanding, inclusion of family in the discussion making process, and relationship within the care situation.” from the method section. These are results and therefore do not belong there. 

Thank you for this comment. We have removed it from the method section. 

3.Please cite the excluded studies (n=9) in the second last step of the PRISMA flow chart. This allows the reader to review these excluded studies. 

We have added the nine excluded studies as supplementary material and referenced in the result section: 

Result section:

These nine publications were excluded, ultimately resulting in four systematic reviews eligible for inclusion. Appendix search strategy

4 - Table 3: Is it possible to provide the total number of participants per included review? A separate column specifying the different outcomes studied would also be helpful. The ‘target population’ is in my opinion the ‘setting’ since the population could be healthcare workers, patients and/or family. 

Thank you for these points. 

We agree that ‘Setting’ is a better name and amended Target population to Setting. 

We have also added a separate column with the different outcomes studied.

We agree with the reviewer that in certain cases it may indeed be opportune and desirable to enumerate samples in the synthesis of systematic reviews. We have also considered this before, but concluded that the included designs differed to such an extent that an addition could give an unsuitable result. In other words, there could even be participants in (semi)qualitative designs that would be added to quantitative designs. To counteract this bias, we have chosen to present only the size of the sample per included study (systematic review) in a table (see table 3).

5 - In a systematic review of systematic reviews summary results should be presented per included systematic review in a summary of findings table. The authors stated that included studies were too heterogeneous in their methodology and outcome measures, making it impossible to perform meta-analyses. And I do agree that this is the case here. However, the authors could decide to report effect sizes and corresponding levels of uncertainty per type of outcome for each category (e.g., family member satisfaction) in a table. If necessary, the different definitions of outcomes could be provided next to the results. Moreover, the authors could decide to select studies with a certain level of evidence (e.g. RCTs and/or quasi-experimental research) and provide a table with the respective effect sizes per outcome measure. I think it would be interesting to present some numbers in this systematic review. 

We thank the reviewer for this important point of view. We have set the suggestion given here as our goal before. We thought we might be able to present clarifying forest plots as well. However, in line with the previous remark, this turned out not to be possible because of the too great chance of bias in the comparison of the samples and the methods used.

6 - p20. The authors state: “However, in the current context of healthcare, with the growing number of care needs and increased complexity of care, the involvement of family caregivers is desirable and necessary. Current healthcare systems are increasingly relying on family members to take on care responsibilities. This is also true for a wide variety of adult patients and not strictly for the most seriously ill, unresponsive ones or those without decision-making capacity [69].” Yes, this is true. However, isn’t an increased reliance on family members because of shortages in professional caregivers a negative evolution? The authors should nuance their message here or at least provide some more explanation. I do not think that the argument is entirely correct, and this statement is repeated in the conclusion (In the light of the current challenges in healthcare, FCRs seem to be highly relevant for adult non-critical care patients as well.) 

Discussion

However, in the current context of healthcare, with the growing number of care needs and increased complexity of care, the involvement of family caregivers is desirable and necessary. Current healthcare systems are increasingly relying on family members to take on care responsibilities. This is also true for a wide variety of adult patients and not strictly for the most seriously ill, unresponsive ones or those without decision-making capacity [69].” 

The sentence is rephrased as:

Family members are the major source of support for patients during hospitalization and at home. Therefore, family involvement is essential to ensure quality and continuity of care. Furthermore, in the current context of healthcare, with the growing number of care needs and increased complexity of care, the involvement of family caregivers becomes even more desirable and necessary. This is true for a wide variety of adult patients and not strictly for the most seriously ill, unresponsive ones or those without decision-making capacity.

Conclusion

In the light of the current challenges in healthcare, FCRs seem to be highly relevant for adult non-critical care patients as well.

Rephrased as:

Since family is the major source of support for patients and current healthcare more and more relies on the support of family, FCRs seem to be highly relevant for adult non-critical care patients as well.

7 - An important limitation of this review is the limited focus on the added value of FCRs. The authors acknowledge this in the discussion and mention that implementation should be studied. However, could this be more substantiated with relevant literature? Furthermore, the organization and education of FCRs are important aspects of implementation and should be studied. 

Amended to: 

Furthermore, the organization and education of FCRs are important aspects of implementation. A structure or protocol with clear roles of FCRs participants should be studied and developed [71,72].

---

## [Editor Report · Decision Letter 2]

21 Dec 2022

The added value of family-centered rounds in the hospital setting: A systematic review of systematic reviews

PONE-D-22-14357R2

Dear Dr. Woldring,

We’re pleased to inform you that your manuscript has been judged scientifically suitable for publication and will be formally accepted for publication once it meets all outstanding technical requirements.

Kind regards,

Inge Roggen, M.D., Ph.D.

Academic Editor

PLOS ONE
---

## [Editor Report · Acceptance letter]

2 Jan 2023

PONE-D-22-14357R2 

The added value of family-centered rounds in the hospital setting: A systematic review of systematic reviews 

Dear Dr. Woldring:

I'm pleased to inform you that your manuscript has been deemed suitable for publication in PLOS ONE. Congratulations! Your manuscript is now with our production department. 

Kind regards, 

on behalf of

Dr. Inge Roggen 

Academic Editor

PLOS ONE